# Detection of Superficial Defects in the Insulation Layers of Aviation Cables by Infrared Thermographic Technique

**Fang Wen** [1,2,*] **and Jie Jin** [1]

1   School of Energy and Power Engineering, Beihang University, Beijing 100190, China; jinjie@buaa.edu.cn
2   AVIC China Aero-Polytechnology Establishment, Beijing 100028, China
*   Correspondence: wfzz314@163.com; Tel.: +86-135-0109-2943

**Abstract:** Superficial defects in the insulation layers of aviation cables can cause serious failures of and disasters for aircraft. Considering the critical importance of safety for aircraft, there is a need to develop a nondestructive technique to detect these kinds of defects in aviation cables. The objective of this paper is to investigate defect detection in aviation cable insulation by an infrared technique. The temperature distribution in the tested cable insulation surface under different thermal excitations and its time variation law are firstly analyzed by numerical simulation. Experimental testing is simultaneously conducted to study the influence of insulation wear defects on the temperature distribution of the cable surface. Complex background temperature distributions are eliminated in infrared images to improve the recognition of defects and extract the difference of the cable surface to draw a curve. The obtained results clearly indicate that the temperature variation interval in the curve can successfully reflect the sizes and locations of insulation defects.

**Keywords:** infrared technique; insulation layers; aviation cable; thermographic image





## 1. Introduction

The electrical wiring interconnection system (EWIS) plays key roles in aircraft, which composes various cables and line devices to transmit electrical energy or signals in aircraft [1]. Ensuring the stability of the EWIS is an important issue for the safety of aircraft. Statistics show that insulation wear accounts for 30% of the flaws in aircraft cables [2,3]. In addition, many accidents and aircraft failures are directly related to the flaws in the cable insulation of the EWIS, such as the American TWA B747 plane in 1996 and the Swissair MD-11 plane in 1998, both which had flight accidents due to flaws in the insulation layer of their aviation cables [4]. Therefore, it is very important to investigate and develop the defect detection technology of the cable insulation layer to ensure the flight safety of civil aircraft.

Current aviation cable detection methods and equipment have relatively weak detection capabilities and poor detection results, which makes it difficult to ensure aircraft flight safety [5–7]. Various nondestructive testing (NDT) methods for damaged cable detection have been reported. The time–domain reflection method of the reflectometer is sensitive for cable surface detection, but the requirement of cable surface cleanliness is high, and it is not suitable for in situ detection [8–10]. Radiographic testing equipment is complex, with a certain radiation hazard [11]. The capacitance detection method can only be used to measure the dielectric constant of a single cable insulation layer [12]. Ultrasonic methods require a high cable surface cleanliness for the use of the couplant [13,14]. The eddy current method is not available for cases wherein the conductor inside the cable is broken, or for the detection of non-conductive insulation layers [15]. Capacitive tomography is susceptible to complex environments in the detection of cable insulation defects [16].

Most of the enterprises follow the traditional manual inspection methods, namely the visual inspection method and the megohmmeter or multimeter measurement methods [17,18]. The visual inspection method uses flashlights, reflectors, magnifying glasses, and other

equipment to check the insulation layer of the cable section for cracks, wear, and other fault detection. The megohmmeter method mainly measures insulation resistance, and the multimeter method generally measures electrical parameters such as resistance, voltage, and current to determine whether each cable is on or off, one by one. The megohmmeter method and the multimeter methods have complicated procedures, low efficiency, and easy-to-miss detection. Some researchers have developed a hand-held aviation cable detector based on the time–domain reflectometry method [19]. By transmitting a pulse signal to one end of the cable, the impedance mismatch characteristics of the fault point are used to obtain the reflected signal parameters of the fault point so as to determine the type of fault and the location of the fault point. This method is a typical single-ended test method which can effectively solve the problem of cable fault detection in "invisible and unreachable" locations on the aircraft. However, the reflectometer is very sensitive, so the cleaning of the cable surface and the connection between the reflectometer and the cable being tested should be very carefully prepared. In addition, this method is not suitable for the in situ detection of the cable during the service of the aircraft. White et al., of Johns Hopkins University in the United States, used three methods (infrared thermal imaging, time domain reflectometry, and pulsed radiography) to detect aviation and aerospace cables, and to detect short circuits in the cable bundles and insulation damage [9]. Li et al. of Iowa State University used an lnductance, capacitance and resisitance (LCR) tester to measure the dielectric constants of polytetrafluorethylene (PTFE) and ethylene-tetrafluoroethylene (ETFE), which are insulating materials for aviation and aerospace cables. The dielectric constants reflect the performance degradation of the insulating materials under ambient temperature changes [20]. Chen et al., from Iowa State University in the United States, designed and produced a capacitance probe for the quantitative non-destructive testing of the insulation performance of aviation cables, and judged whether the cable had insulation aging defects through the change of complex permittivity [10].

Infrared detection technology is suitable for the rapid in situ detection of a large structure, and it plays an increasingly important role in fault detection in power systems, equipment, and cables [17]. Sfarra et al. from Italy systematically investigated the infrared thermographic technique for defect detection in both numerical and experimental manners [21,22]. In this paper, infrared detection technology was used to detect the insulation layer of the EWIS cable. A halogen lamp was used to heat the cable. Temperature distribution on the surface of the cable was obtained by an infrared thermal imager, and then the integrity of the cable insulation layer was analyzed.

## 2. Numerical Simulation

Earlier investigations showed that numerical simulation can eliminate the influence of uneven heating, environmental interference, and other factors on the surface temperature of the cable [23], which is conducive to the analysis of the heat transfer phenomenon of the cable and provides a theoretical basis for the experimental analysis. Commercial finite element analysis software was used to calculate the surface temperature field of the cable insulation under the halogen lamp heating, and the influence of insulation defects on the surface temperature field was analyzed.

### 2.1. Numerical Model

The aviation cable model was firstly established. The material parameters of the cable are shown in Table 1. In the model, the material of the cable insulation layer is set to PTFE, the thickness of the insulation layer is 0.23 mm, and the cable core consists of 7 conductors. The conductor material is copper, and the radius of each conductor is 0.102 mm. The insulation layer was cut with an ellipsoid at the axial middle position of the cable model to simulate cable wear. In the longitudinal section of the cable, the ellipse defect was made with its major semi-axis (half of flaw length) being 1.0 mm, and its minor semi-axes (flaw depths) set to 0.2, 0.15, 0.1, and 0.05 mm, respectively. In the numerical simulation, we employed an external radiation source to heat the cables to simulate the halogen lamp. The

heat radiation direction was 45 degrees compared to the normal direction to the location of the flaw. Figure 1 shows the geometry and dimensions of the cable model (with an insulation flaw of 0.2 mm depth).

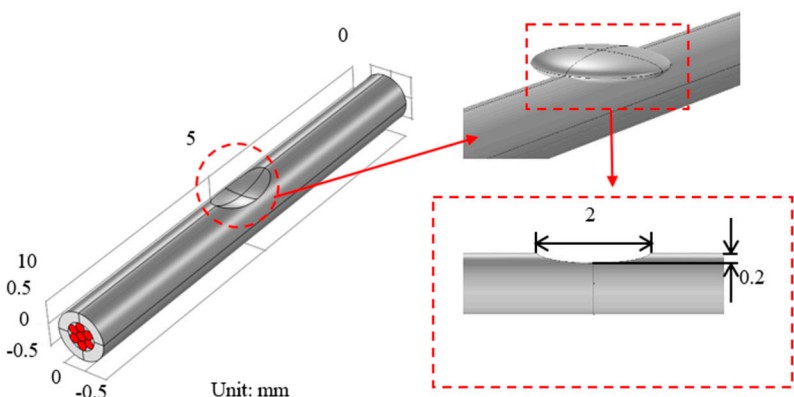

**Figure 1.** Schemes of the tested sample with the flaw in the numerical model.

**Table 1.** Cable model material parameters.

| Material | Attribute | Value |
|---|---|---|
| Copper | Density | 8960 kg/m$^3$ |
| | Thermal conductivity | 400 W/(mK) |
| | Heat capacity at constant pressure | 385 J/(kg·K) |
| PTFE | Density | 2200 kg/m$^3$ |
| | Thermal conductivity | 0.24 W/(mK) |
| | Heat capacity at constant pressure | 1050 J/(kg·K) |

### 2.2. Simulation of Infrared Detection of Cables

We firstly set up the heat transfer module and load boundary conditions and simulated natural convection between the cables and air. The power of the external radiation source was set to 1000 W, and the cable was heated by simulating a halogen lamp. The heat radiation direction was directly opposite to the location of the flaw. The heating time was 300 s. At that time, the surface temperature of the cable was stable.

Figure 2 is a three-dimensional distribution diagram of the surface temperature of the cable, with a flaw depth of 0.2 mm in the insulating layer, at the end of the heating time. As can be seen in Figure 2, the temperature at the cable flaw was significantly lower than the temperature at the nearby non-flaw. The halogen lamp radiated heat to the surface of the cable. Since the insulation layer at the flaw was thin, heat was transferred from the flaw surface to the conductor faster, and the conductor had a strong thermal conductivity and the heat was rapidly dispersed along the conductor. Therefore, the surface temperature of the flaw was lower than the flaw-free surface.

The influence of the defect depth of the insulation layer on the surface temperature of the cable was further analyzed. The depth of the wear flaw of the different insulation layers was set and calculated. Figure 3 shows the temperature distribution of the cable surface corresponding to the depth of the flaw. When the flaw depths are 0.05, 0.1, 0.15, and 0.2 mm, the center temperature at the bottom of the flaw is 37.96, 37.87, 37.74, and 37.53 °C, respectively. The deeper the flaw, the lower the center temperature at the bottom of the flaw. Meanwhile, when the cable is defective, the surface temperature of the cable outside the flaw is higher than the surface temperature of the non-defective cable. This is because when there is a flaw, the heat is conducted from the flaw surface to the conductor faster, the conductor has a strong thermal conductivity, and the temperature of the entire conductor rises rapidly, making the temperature difference between the inside of the cable outside the flaw and the surface of the cable smaller, slowing down the outside of the flaw. The speed

of heat transfer from the surface of the cable to the inside of the cable eventually leads to a high temperature on the surface of the cable outside the flaw. The deeper the flaw, the higher the surface temperature of the cable outside the flaw.

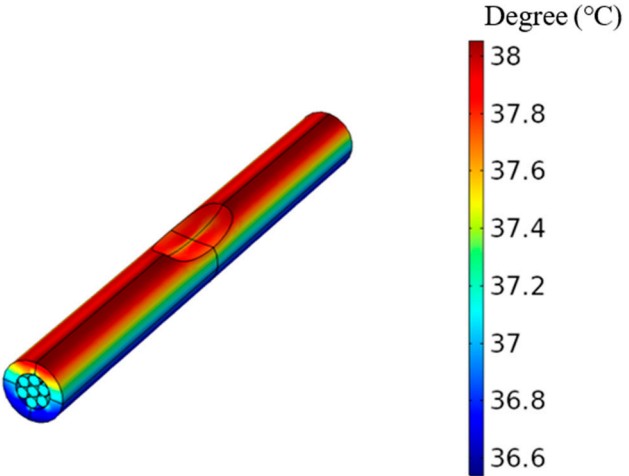

**Figure 2.** 3D distribution of cable surface temperature (heating time of 300 s).

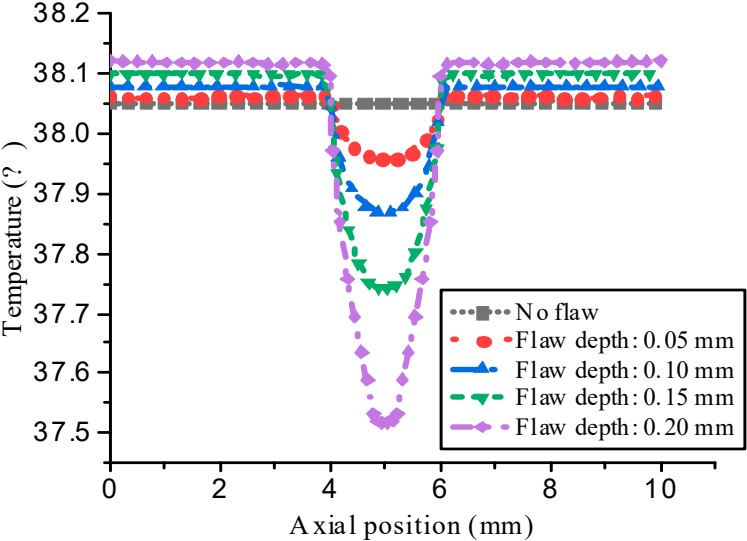

**Figure 3.** Temperature distribution of the cable surface with different defect depths.

## 3. Experiment

### 3.1. Experimental System and Sample

The infrared nondestructive testing system is shown in Figure 4. The external thermal excitation source (halogen lamp) heated the tested cable sample, and the heat flowed inside the cable, breaking the original thermal equilibrium state, and the infrared thermal imager obtained the cable. The sequence of surface heatmaps, through the analysis of the sequence of heatmaps, realized the identification of defects in the cable insulation layer. The halogen lamp power was 1000 W, which was set about 25 cm away from cable. The infrared thermal camera was a FLUKE TI200, which has a horizontal FOV of 45° (approximately), 160 pixels horizontally, and an IR lens of focal length 6.5 mm. The official IFOV from the specs sheet is 5.2 mrad, and a focus distance of 30 cm was used. The infrared spectral band was 7.5~14 μm. During the experiment, the infrared thermal imager was placed 30 cm away from the cable. Thermal image resolution, or spatial resolution, is an important parameter, e.g., field of view (FOV), instantaneous field of view (IFOV), and detector array, which were considered when choosing the infrared camera. These parameters can be used to indicate

the ability of the camera to distinguish between two objects in the field of view, which primarily depends on the object-to-camera distance, lens system, and detector size [24–27]. The camera lens can be carefully adjusted to achieve a better image effect.

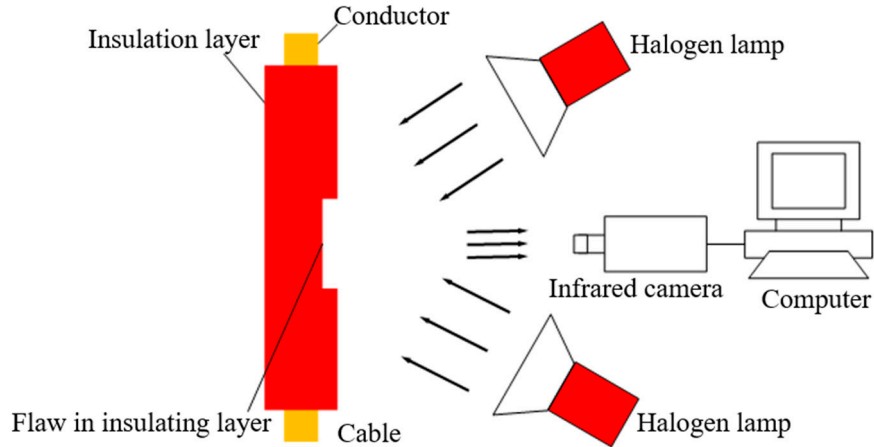

**Figure 4.** Experimental setup of the infrared testing system.

The cable sample model of the first group is MIL-W-22759/35-24, and the parameters are shown in Table 2. The measured objects are the whole insulation cables with lengths of 45 cm and different diameters (Group #1 and Group #2). Artificial defects were made on the cables to simulate wear defects. The article defects were made in the surface of the cables. The specific values are shown in Table 3. In order to measure the length of the defect, we fixed the five cables with a foam box and took the third cable as a reference cable.

**Table 2.** Parameters of the Group #1 tested cable.

| Characteristic | Value |
| --- | --- |
| Number of conductors | 19 |
| Material of insulating layer | XL-ETFE |
| Conductor material | Silver-plated high-strength copper alloy |
| Outer diameter of conductor (mm) | 0.572~0.620 |
| Outer diameter of cable (mm) | 1.092~1.194 |
| Color | White |

**Table 3.** Parameters of the defects in the insulation layer of the Group #1 cable.

| Cable Number | Length of Flaw (mm) | Depth of Flaw (mm) |
| --- | --- | --- |
| 1 | 4 | 0.115 |
| 2 | 7 | 0.23 |
| 3 | / | No |
| 4 | 7 | 0.0575 |
| 5 | 4.5 | 0.23 |

The cable samples used in the second set of experiments were collected from a retired Boeing 747. The cables were routed separately. We checked the integrity of the tested samples. Before introducing the defects into the cables, we cleaned their surfaces and checked their integrity visually. The cable parameters are shown in Table 4. Two defects were artificially created on the cables to simulate wear defects, and the lengths of the defects were measured by a ruler. The specific values are shown in Table 5.

**Table 4.** Parameters of the Group #2 cables.

| Characteristic | Value |
| --- | --- |
| Number of conductors | 7 |
| Material of insulating layer | PTFE |
| Conductor material | Nickel-plated high-strength copper alloy |
| Conductor diameter (mm) | 0.78 |
| Thickness of insulating layer (mm) | 1.67 |
| Color | White |

**Table 5.** Parameters of the defects in the insulation layer of the Group #2 cables.

| Location of the Defect | Length of Defect (mm) | Depth of Flaw |
| --- | --- | --- |
| Left | 12 | 1/2 the thickness of the insulating layer |
| Right | 12 | Thickness of the entire insulating layer |

*3.2. Initial Test*

In this study, we recorded original heatmap for every time period of heating and then subtracted respective heatmaps from the thermograms with cables. We fixed the insulation cables with different defects inside an open box to ensure a consistent background of all the cables. Secondly, we processed the infrared images to minimize the interference of the complicated background temperature distribution. In this study, the temperature distribution on the tested samples with different heating times were firstly observed and analyzed for optimizing relative parameters to obtain the desired results.

The samples of Group #1 were heated for 80 s, and the distribution of the surface temperature of the cables when heated for 10 and 80 s was obtained by using an infrared thermal imager. The power of the halogen lamp was turned off to stop heating. An infrared thermal imager was used to obtain the distribution of the surface temperature of the cable when the cable was naturally cooled for 20 and 80 s. The values of time for heating and cooling were selected by referencing the manual introduction of the halogen lamp. In addition, we also compared the infrared images during the heating and cooling process. It was noticed that at these four different times, we could obtain the obvious differences of image features.

Figure 5 shows the temperature distribution of the cable surface at the four time points, respectively. After heating for 10 s, the temperature at the defect of the second cable and the fifth cable was significantly lower than the temperature at the defect-free place (Figure 5a), indicating that the use of infrared detection technology could effectively detect the insulation layer of the cable defect. After heating for 80 s, the display of defects was more obvious (Figure 5b). In the cooling stage, with the increase of cooling time, the temperature difference between the defect and the non-defect on the surface of the cable decreased.

It is important to note that the insulation cables of aircraft are generally placed in a limited space inside of the aircraft structure, without a relative ideal background. In addition, one of the objectives of this study was to study and remove the complicated background images for the improvement of infrared images of defects. Thus, we did not perform the testing in front of a cold wall with a high thermal mass.

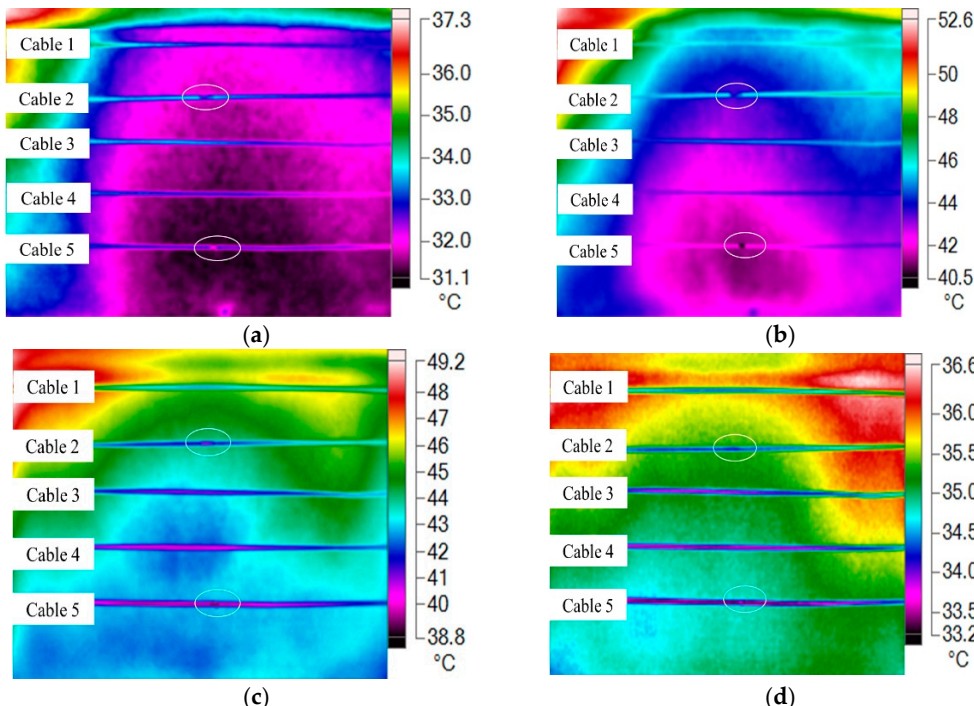

**Figure 5.** Surface temperature distribution of the Group #1 cables. (**a**) Heating up for 10 s. (**b**) Heating up for 80 s. (**c**) Cooling off for 20 s. (**d**) Cooling off for 80 s.

The cable samples of Group #2 were heated and the temperature distribution of the cable surfaces was also obtained. The cable with the defect on the front was heated for 120 s, and the temperature distribution of the cable surface for the different heating times was obtained, as shown in Figure 6. In the first 100 s heating process, both full-wear defects and half-wear defects were displayed. The full-wear defects were more obvious than the half-wear defects. When heated for 120 s, almost no defects were observed from the heatmap. We heated the cable with the defect on the reverse side for 110 s to obtain a heatmap sequence, as shown in Figure 7. During the first 90 s heating process, full-wear defects and half-wear defects were also displayed, and after 110 s of heating, almost no defects could be observed from the heatmap. It is important to note that we intentionally did not set the non-focusing thermal irradiation to explore the influences on defect detection in this study, since in some practical testing, it cannot always ensure the thermal focusing. To improve the accuracy of defect identification, the infrared images were processed to reduce the interference of the background temperature. 1

Based on the comparisons, it was found that the defect could be clearly identified either in the front or back sides of the cables when we did not heat the cables for a long time period (in this study it was about 100 s). Thus, the obtained results suggest that defects existed in different positions on the cables could be identified with similar effects as in the case of the cables not being heated for a long time. It is important to note that we did not intentionally set the focusing condition to achieve the infrared images with a high quality. The current investigation is still working on the edge of focal distance for the significant effect of thermal focusing on infrared measurements.

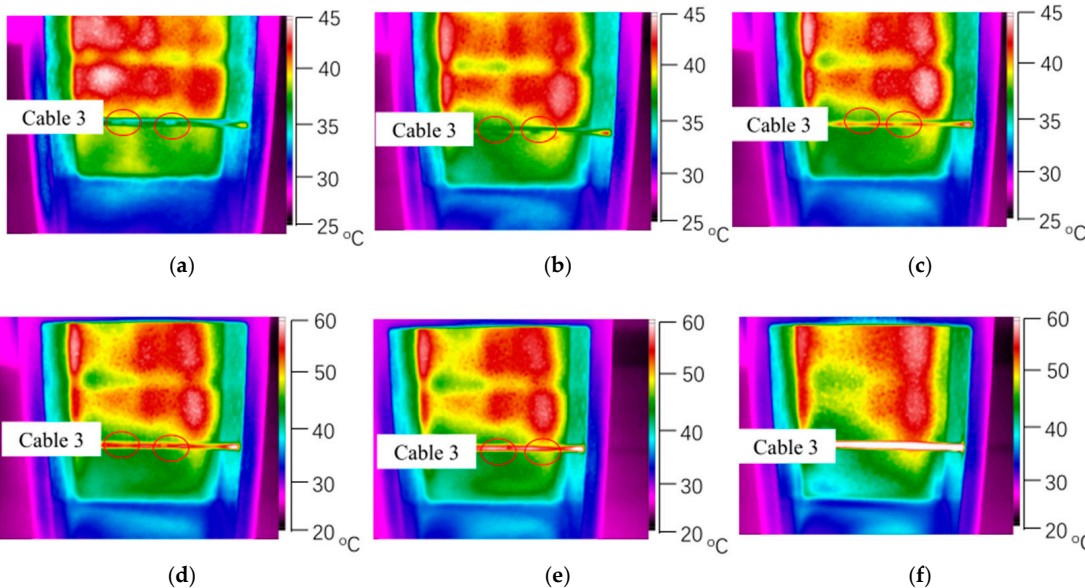

**Figure 6.** Surface temperature distribution of the Group #2 cables at different times during the heating phase (defects are on the front of the cable). (**a**) Heating up for 20 s. (**b**) Heating up for 40 s. (**c**) Heating up for 60 s. (**d**) Heating up for 80 s. (**e**) Heating up for 100 s. (**f**) Heating up for 120 s.

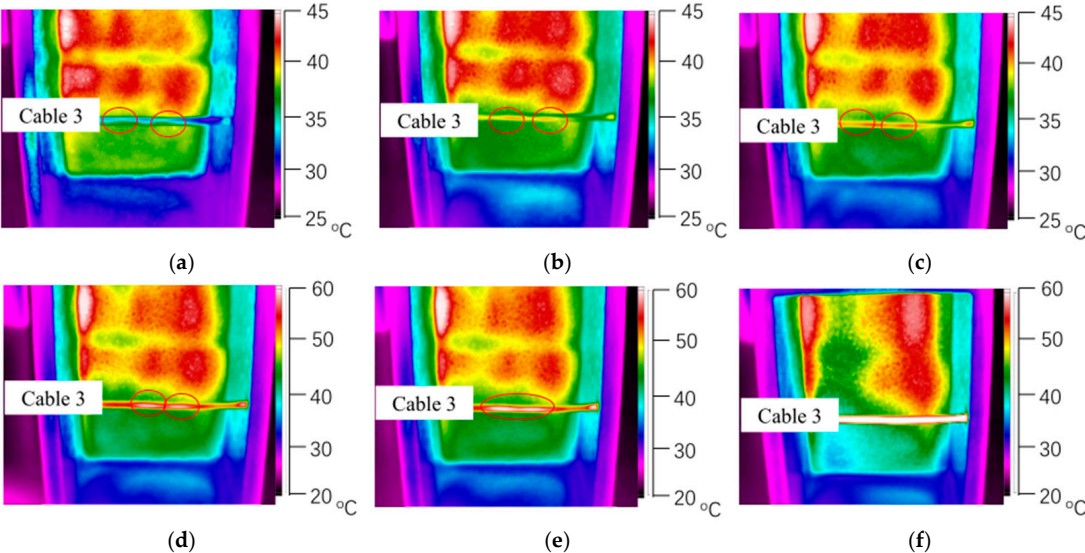

**Figure 7.** Surface temperature distribution of the Group #2 cables at different times during the heating phase (the defect is on the back of the cable). (**a**) Heating up for 20 s. (**b**) Heating up for 40 s. (**c**) Heating up for 60 s. (**d**) Heating up for 80 s. (**e**) Heating up for 100 s. (**f**) Heating up for 120 s.

## 4. Infrared Image Processing and Analysis

The infrared images obtained by the infrared thermal imager contain the interference of the background temperature with the complex distribution [28]. In order to improve the accuracy of the defect identification, the infrared image was processed by commercial software. The linear interpolation method was used to obtain the background temperature data in the cable area from the background temperature data outside the cable area. We constructed the background temperature image and subtracted the original heat map from the background temperature image to obtain the differential image. The accuracy of the defect identification was improved by reducing the interference of the background temperature. Finally, the data of the cable area was extracted from the differential image to

draw a curve to analyze the effect of the defects on the surface temperature of the cable. The infrared image processing flow is shown in Figure 8.

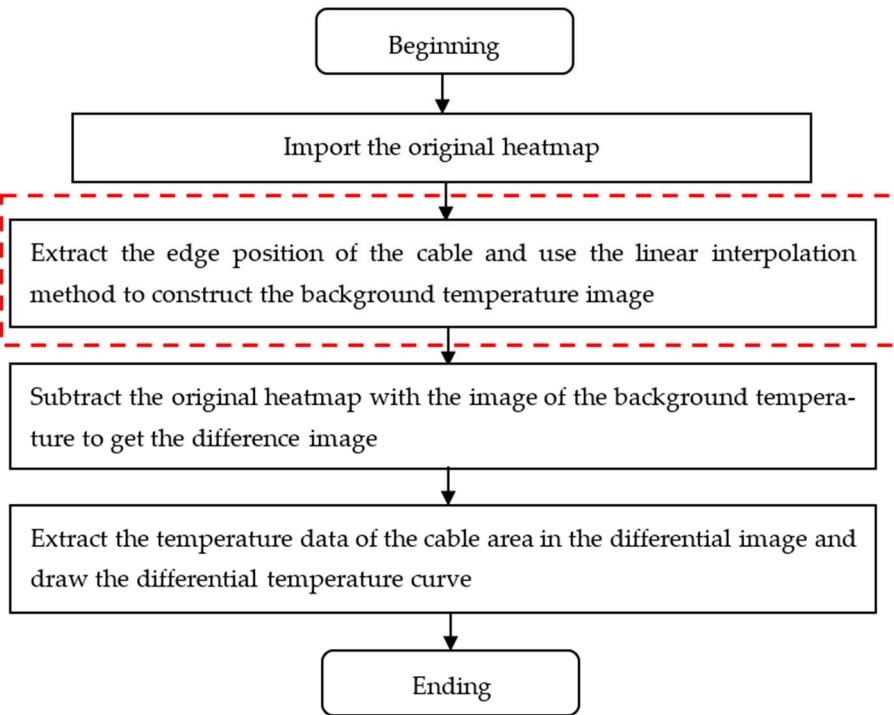

**Figure 8.** Flowchart of the infrared image processing for cable detection.

The infrared images of the first group of cables heated for 80 s were taken as an example for image processing. The original heatmap is shown in Figure 9a. The position of the edge of the cable area was extracted. The temperature data in the cable area was obtained from the temperature data outside the cable area through the linear interpolation method to construct a background temperature image, as shown in Figure 9b. The original heat map was subtracted from the background temperature map to obtain a differential image without the background temperature, as shown in Figure 9c. The results of image after processing show that the defects are more obvious in the image after removing the interference of the background temperature.

However, although the temperature change range at the cable defect in the heatmap is related to the length of the defect, the length of the defect cannot be effectively obtained by only observing the heat map and the differential image. Therefore, a signal processing program was used to extract the cable in the differential image (Figure 9c). After obtaining Figure 9c, we used commercial software (Matlab) to extract the image features, including the three sets of differential temperature data in the cable area in the differential images. Then, the three sets of differential temperature data at the center of the cable could be achieved by using those at the edge position of the cables. The averages of the three differential temperature data were calculated, and the differential temperature curve was plotted for analysis, as shown in Figure 10. It was found that the length of the defect on the second cable and fifth cable was exactly equal to half of the number of pixels in the temperature anomaly range. The defect lengths of the second and fifth cables were 7.0 and 4.5 mm, respectively. Therefore, by counting the number of pixels in the temperature change area, the length of the defect could be obtained. When the insulation was worn but the conductor was not exposed (as in the first and fourth cables), no defect was observed in the heatmap. However, the temperature change at the defect could be observed in the differential temperature curve.

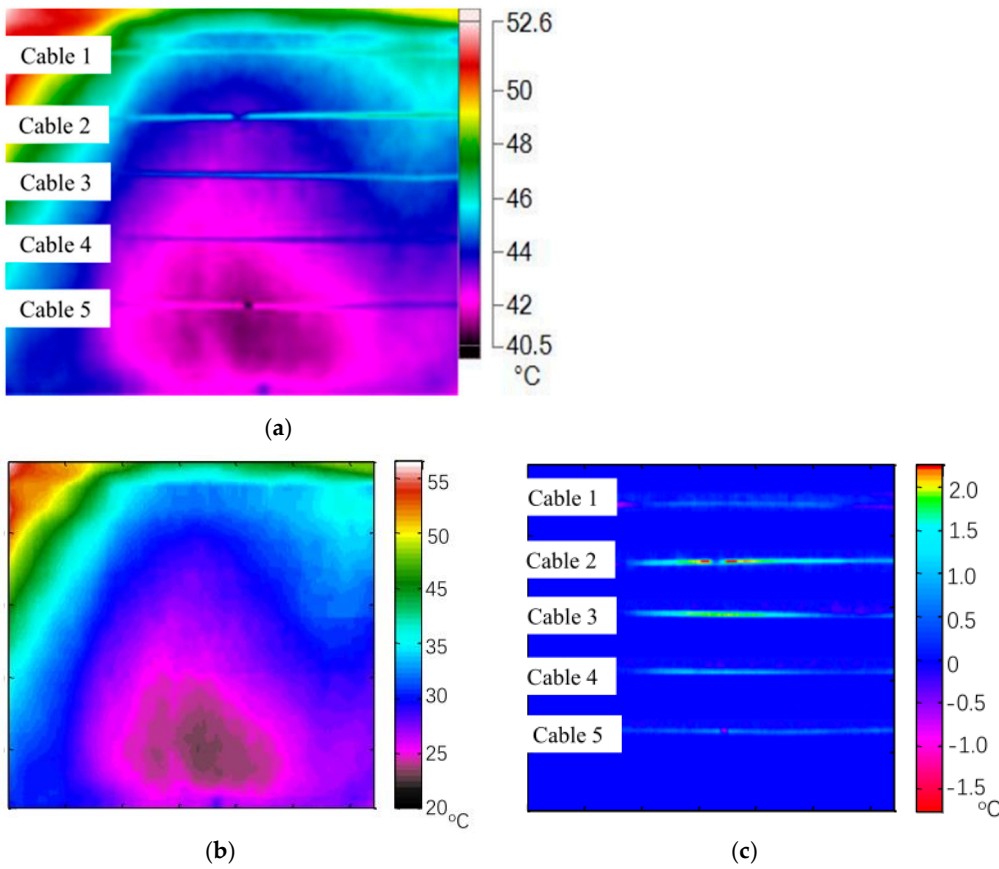

**Figure 9.** Removal of the background temperature of the infrared image (the picture shows the Group #1 cables heated for 80 s). (**a**) Original image. (**b**) Image of background. (**c**) Difference image obtained after subtraction.

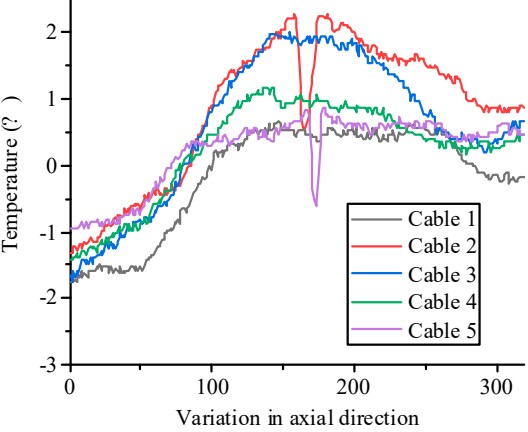

**Figure 10.** Temperature curve of the cable surface (the picture shows the Group #1 cables heated for 80 s).

It should be noted that the numerical study provides us a qualitative suggestion about temperature distribution on the surface of the cables, which can guide the experiment setup. There are some differences between experiment and numerical model in terms of measurement setup or excitation time for the reason that it has an ideal background and heat dissipation from the cables to the air in the numerical simulation. However, in the experiment, the ideal background and uneven heat dissipation caused the obvious differences of the infrared detection results. Thus, we set the different heating times to obtain infrared images in the numerical and experimental studies. It is important to note

that the numerical results are consistent with the experimental ones. Both of them indicate that a larger depth of flow corresponds to a lower surface temperature of a cable with a flaw.

## 5. Conclusions

In this paper, a numerical simulation and an experiment of the infrared detection of defects in the insulation layer of aviation cables was carried out. The cables were heated by means of thermal excitation using halogen lamps. The distribution of the surface temperature of the cables was analyzed. The model was established using commercial software. The corresponding parameters were set for simulation. The simulation results showed that the surface temperature at the defect of the aviation cable was significantly lower than the surface temperature at the defect-free place in the heating stage. This indicated that the depth of the defect corresponds to a lower surface temperature at the defect, which had certain guiding significance for the subsequent experimental research. The experimental results showed that the temperature of the defect on the surface of the cable was lower than that of the non-defect under the action of the halogen lamp's thermal excitation. The infrared inspection of thicker aviation cables illustrates that defects can also be captured by infrared cameras when they are on the reverse side of the cable. The experimental results are consistent with the simulation ones. Both of them indicate that a larger depth of flow corresponds to a lower surface temperature of a cable with a flaw. In addition, the background temperature distribution of the heatmap was reconstructed by linear interpolation. The interference of the background temperature was removed from the heatmap. The results of image processing showed that the defects were more clearly displayed after removing the background temperature interference.

**Author Contributions:** Conceptualization, F.W. and J.J.; methodology, F.W.; software, F.W.; validation, F.W. and J.J.; formal analysis, F.W. and J.J.; investigation, F.W.; resources, J.J.; data curation, F.W.; writing—original draft preparation, F.W.; writing—review and editing, F.W. and J.J.; visualization, F.W. and J.J.; supervision, J.J.; project administration, F.W. and J.J.; funding acquisition, F.W. All authors have read and agreed to the published version of the manuscript.

**Funding:** The work was supported by the Civil Aircraft project of the Ministry of Industry and Information Technology (Grant No. MJ-2018-J-75).

**Institutional Review Board Statement:** Not applicable.

**Informed Consent Statement:** Not applicable.

**Data Availability Statement:** Not applicable.

**Conflicts of Interest:** The authors declare no conflict of interest.

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
