# Peer review of "Detection of Superficial Defects in the Insulation Layers of Aviation Cables by Infrared Thermographic Technique"

_coatings, doi:10.3390/coatings12060745_

Round 1

Reviewer 1 Report

  • Table 1 – introduce spaces between values and units and use superscripts
  • Figure 1 - please take look at the title, I think it is wrong.
  • Row 108 and 145 – the distance of the halogen lamp to the cable is also of importance.
  • Table 3 – please insert depth of flaw units
  • What is the surface condition of the cables of group #2? Since they are taken from the retired aircraft it could happen that the surface is not clean, which would influence the measurements and any impurities on the surface would lead to the false positive conclusions regarding the defect detection.
  • Please elaborate what is the IFOV of the IR camera used and FOV of the setup shown in the figure 4. Also give information on the spatial resolution of the IR camera as well as the lens angle used.
  • Please elaborate how IR camera is taking measurements. I’m asking this since a lot of cameras average 9 pixels to calculate the temperature which is then displayed as i.e. spot temperature.
  • If You are doing IR measurements of such thin cables, aspects like IFOV and the measurement principles related to camera optics should be carefully considered, otherwise You will get more information about the background than cables. This is particularly of issue when the background is unevenly heated while cables are directly compared.
  • Figure 5 – please enlarge the thermograms since encircled defects are not visible in any thermograms except thermogram b.
  • Figure 6 and figure 7 – thermograms are not properly focused
  • Figures 6 and 7 – what were the temperature span for all these thermograms? It seems to me that better thermal focusing would enhance defect detection.
  • Chapter 4 – please explain how the original heatmap was recorded, especially having in mind that the halogen lamps were heating up the background. Did You record original heatmap for every time period of heating (i.e. 10 s, 40 s, 60 s, etc…) and then subtract respective heatmaps from the thermograms with cables? I’m asking this since the background is also getting hotter with longer exposure to the halogen lamps.
  • Please explain why the experiment wasn’t performed in front of the i.e. cold wall with high thermal mass. Additionally, if You have such a background which if relatively far away from the halogen lamps, they would not affect the background making the IR measurements much easier.
  • What are the temperature spans in figures 9 b and 9 c? Please insert the temperature scales.
  • Figure 10 – units are missing on the Temperature axis
  • Please explain how did You extract the temperatures shown in figure 10
  • Please cross-reference the cables shown in figure 9 with the legend in figure 10.
  • Please explain how other three sides of the same cable should be inspected in case that the defect are positioned on the other sides of the cable.
  • Please insert more literature review, especially in the last 5 years.

Author Response

Dear reviewer:

Thanks very much for your constructive suggestions and comments. Clearly, the comments and suggestions are very useful to enhance the quality of the present manuscript. Here, we tried to incorporate them as much as possible in the revised manuscript.

Revised text and a response letter which addresses each point and indicates how the manuscript has been revised are provided.

Reviewer 2 Report

The authors declare that this paper analyses subsurface defects in the insulation layers of aviation cables. In the aeronautics scenario, this typology of defects can be causing a serious failures and disasters of aircraft. For these problems it’s most important to evidence a defect with a non-destructive detection for example by infrared technique. It’s evident that, different current and voltage excitation on the cable, causes a different thermal excitation of the insulation layers. In the other hand for the same electrical excitation, it’s possible you can have a different superficial thermal scenario strictly depend on the conductive/convective dissipation on electrical insulation surface. Experimental testing is simultaneously conducted to study the influence of insulation wear defects on the temperature distribution of cable surface. Following an attentive re-reading on the paper, I invite the authors to modify the word “sub-surface defect” with “superficial defect” for entire article.  

  1. Introduction:

In this section, the authors explain the different methods using in university domain for estimate a defect with different non-destructive techniques. For provide a complete scenario to the reader, please describe -in very quickly mode- what are the non-destructive technique using for testing damage cables by a whatever important aircraft manufacturer. 

  1. Numerical simulation:

Page 2 line 80 - 81 → Please describe, how it was realised the artificial ellipsoid defect for to ensure a correct geometry and very small dimension.

Page 3 line 107 – 108 → Please describe, how is oriented the halogen lamp respect to the defect. If possible, define all the position and orientation cable/lamp whit a schematic figure. Ultimately, I suggest inserting the references

  • Perilli, S., Palumbo, D., Sfarra, S., Galietti, U.

Advanced insulation materials for facades: Analyzing detachments using numerical simulations and infrared thermography

(2021) Energies, 14, 7546. https://doi.org/10.3390/en14227546

Page 4 line 124 → Please verifying a measure: is 0.05m? or 0.05mm?

  1. Experiment:

Page 4 line 140 → Please, takes the place Fig.4 with a figure suggested on Page 3 line 107 – 108. For realize a robust numerical model, the geometry and physical conditions must be identical with a real experiment. For this reason, I suggest to abolish Fig.4 in EXPERIMENT SECTION and using a schematic figure in NUMERICAL SIMULATION SECTION.

Page 4 line 153 → The authors declare than the cables come with a retired Boeing 747. Has been the authors verified the integrity of material?  If so, how?

Page 6 Fig.5 → On the sub-figures a), b), c), d) the palette is stick close to thermogram. This makes a difficult analyse the behaviour. Chosen a range palette -1 Celsius degree- for a specific case, results inappropriate. Given the size of the electrical insulation, I suggest use a range for 0.2 Celsius degree. All images have 5 cables analysed. It’s most important magnification every cable and mark each cable -with a number- for any sub-figure. Moreover, it is deemed impossible to the reader to compare the contribution of the thermography results with those of the numerical simulation. I suggest generating false colour images with Comsol Multiphysics computer programs to estimate the behaviour of the thermographic analysis with those of the numerical simulation. In this regard, I recommend to see the following works and I suggest inserting the references:

  • Sfarra, S., Cicone, A., Yousefi, B., Ibarra-Castanedo, C., Perilli, S., Maldague, X.

Improving the detection of thermal bridges in buildings via on-site infrared thermography: The potentialities of innovative mathematical tools

(2019) Energy and Buildings, 182, pp. 159-171

In the other hand, it’s necessary to the readers, for appreciate a thermal scenario, reporting the cooling images for every 20 s time instant, that is: Cool off for 20s, 40s, 60s, 80s. Lastly, please describe in detail all sub-figure.

Page 7 Fig. 6 → On the sub-figures a), b), c), d), e), f) there aren’t palette. Please insert a palette. Given the size of the electrical insulation, I suggest use a range for 0.2 Celsius degree. All images have 1 cable analysed. It’s most important magnification the cable. Moreover, it is deemed impossible to the reader to compare the contribution of the thermography results with those of the numerical simulation. I suggest generating false colour images with Comsol Multiphysics computer programs to estimate the behaviour of the thermographic analysis with those of the numerical simulation. Lastly, please describe in detail all sub-figure.

Page 7 Fig. 7 → On the sub-figures a), b), c), d), e), f) there aren’t palette. Please insert a palette. Given the size of the electrical insulation, I suggest use a range for 0.2 Celsius degree. Why is the time spacing on the sub-figures is not homogeneous? Moreover, it is deemed impossible to the reader to compare the contribution of the thermography results with those of the numerical simulation. I suggest generating false colour images with Comsol Multiphysics computer programs to estimate the behaviour of the thermographic analysis with those of the numerical simulation. Lastly, please describe in detail all sub-figure.

  1. Infrared images processing and analysis

Page 7 line 198 – 207 → The description of the elaboration of the authors is more like a manual for technical use of thermography than to a scientific approach. The authors are requested to treat in the description the typical problems of the IFOV and the FOV giving a scientific approach to the problem. Furthermore, there are many papers in the literature on image processing, among many e.g. (Ibarra-Castanedo et al., Maldague et all. etc.). Authors are requested to read up on these papers and to accurately describe the processing analysis.

Page 8 Fig. 8 → This flowchart is it an innovative concept for image processing or already exists in literature? If it already existed, the authors are asked to add a reference.

Page 9 Fig.9 → On the sub-figure a) the palette is stick close to thermogram this makes a difficult analyse the behaviour and the sub-figures b), c), there aren’t palette. Please insert a palette. Chosen a range palette -2 Celsius degree- for a specific case, results inappropriate. Given the size of the electrical insulation, I suggest use a range for 0.2 Celsius degree. All images have 5 cables analysed. It’s most important magnification every cable and mark each cable -with a number- for the necessary sub-figure. Moreover, it is deemed impossible to the reader to compare the contribution of the thermography results with those of the numerical simulation. I suggest generating false colour images with Comsol Multiphysics computer programs to estimate the behaviour of the thermographic analysis with those of the numerical simulation.

Page 9 Fig.10 → The temperature axis in figure is normalized? Or the zero-point temperature is it centred at specific value choose with respect to an “particular” behaviour of the system? Please, describe the choose the authors and define the measure unit. It’s impossible, for the readers, estimated the temperature behaviour reported in fig. 10 with the temperature behaviour obtained with thermography technique. Furthermore, it's impossible to compare the previous results with the numerical simulation because reported in different numerical scale. Please report all the figures it the same scale for permit to readers to compare the results.

  1. Conclusion

Please, verify that the existing conclusion is close to the new comparative analysis obtained through correspondence for all the results in a new according to scale.

Author Response

(The authors gave the same response as above.)

Reviewer 3 Report

The manuscript presents an application of step heating thermography for the search for subsurface defects in aviation cables. 

The idea seems good and in the introduction many techniques are presented but the inspection of the cables, but little is said about thermography and the state of the art of it where you can find examples similar to this in other applications. 

The state of the art must be completed with references about step heating thermography.

The manuscript presents a simulation using COMSOL software to show how the heating and surface temperature in the cable varies. And then it makes some practical measures on two groups of cables. 

The data used in the simulation, such as physical measurements, excitation time are not compared with the experimental data where they use others especially in the heating time that goes from being 300s in simulation to 80s in experiments without explaining why.

Figure 5,6, 7 is not clear what it represents. One cable, all cables? A photo of a measured cable is missed to see its appearance next to these figures.

It is not explained why they are measured in only 4 im´sgenes and why in those values of time. A complete sequence of thermogram could have been taken during heating and cooling and take advantage of the temporal surface temperature information to explain that those values of time might be optimal.

In summary, the manuscript must be extensively revised so that the theoretical relationship (simulation) with the practical measures is clearly established. Explain why only 4 thermograms are taken and why in those temporary moments. 

The number of references should be increased with publications of similar applications of thermography. And finally a comparative theory-experiment must be included.

Author Response

(The authors gave the same response as above.)

Round 2

Reviewer 1 Report

Please find comments  marked red in the attached document

Author Response

Dear reviewer:

Thanks very much for your constructive suggestions and comments. Clearly, the comments and suggestions are very useful to enhance the quality of the present manuscript. Here, we tried to incorporate them as much as possible in the revised manuscript.

Revised text and a response letter which addresses each point and indicates how the manuscript has been revised are provided.

Sincerely yours!

Fang Wen

Reviewer 2 Report

I thank the authors for the corrections made. The manuscript can be published as is

Author Response

Dear reviewer:

Thanks for your positive comment and suggest the paper to be accepted as it is.

Fang Wen
